

# Diatoms as indicators of the effects of river impoundment at multiple spatial scales

Hendrik J. Krajenbrink[1,2,6], Mike Acreman[3], Michael J. Dunbar[4], Libby Greenway[5], David M. Hannah[2], Cédric L.R. Laizé[2,3], David B. Ryves[1] and Paul J. Wood[1]

[1] Geography and Environment, Loughborough University, Loughborough, United Kingdom
[2] School of Geography, Earth and Environmental Sciences, University of Birmingham, Birmingham, United Kingdom
[3] Centre for Ecology and Hydrology, Wallingford, United Kingdom
[4] Environment Agency of England, Exeter, United Kingdom
[5] Environment Agency of England, Tewkesbury, United Kingdom
[6] Current affiliation: KWR Water Research Institute, Nieuwegein, The Netherlands

Corresponding author
Hendrik J. Krajenbrink,
h.j.krajenbrink@lboro.ac.uk

## ABSTRACT

River impoundment constitutes one of the most important anthropogenic impacts on the World's rivers. An increasing number of studies have tried to quantify the effects of river impoundment on riverine ecosystems over the past two decades, often focusing on the effects of individual large reservoirs. This study is one of the first to use a large-scale, multi-year diatom dataset from a routine biomonitoring network to analyse sample sites downstream of a large number of water supply reservoirs ($n = 77$) and to compare them with paired unregulated control sites. We analysed benthic diatom assemblage structure and a set of derived indices, including ecological guilds, in tandem with multiple spatio-temporal variables to disclose patterns of ecological responses to reservoirs beyond the site-specific scale. Diatom assemblage structure at sites downstream of water supply reservoirs was significantly different to control sites, with the effect being most evident at the regional scale. We found that regional influences were important drivers of differences in assemblage structure at the national scale, although this effect was weaker at downstream sites, indicating the homogenising effect of river impoundment on diatom assemblages. Sites downstream of reservoirs typically exhibited a higher taxonomic richness, with the strongest increases found within the motile guild. In addition, Trophic Diatom Index (TDI) values were typically higher at downstream sites. Water quality gradients appeared to be an important driver of diatom assemblages, but the influence of other abiotic factors could not be ruled out and should be investigated further. Our results demonstrate the value of diatom assemblage data from national-scale biomonitoring networks to detect the effects of water supply reservoirs on instream communities at large spatial scales. This information may assist water resource managers with the future implementation of mitigation measures such as setting environmental flow targets.

## INTRODUCTION

The construction of dams and creation of reservoirs, also known as river impoundment, is among the most fundamental anthropogenic changes to rivers internationally (*Malmqvist & Rundle, 2002*; *Zarfl et al., 2015*). Around 59,000 large dams (higher than 15 m or impounding more than three million cubic metres) exist today (*ICOLD, 2018*), interrupting free-flowing rivers globally (*Stanford & Ward, 2001*; *Nilsson et al., 2005*). There is an extensive literature on the first-, second- and third-order effects of river impoundment (sensu *Petts, 1984*), including changes to sediment transport and channel morphology (e.g., *Sear, 1995*; *Yang et al., 2011*), water quality (e.g., *Ahearn, Sheibley & Dahlgren, 2005*; *Casado et al., 2013*), and flow regime (e.g., *Higgs & Petts, 1988*; *Fitzhugh & Vogel, 2011*). Research quantifying the effects of river impoundment on biotic communities have predominantly concentrated on macroinvertebrate (e.g., *Jackson, Gibbins & Soulsby, 2007*; *Schneider et al., 2018*) and fish communities (e.g., *Cooper et al., 2016*; *Oliveira et al., 2018*).

There is a vast and wide-ranging literature on the use of diatoms to characterise environmental conditions, predominantly within the fields of palaeoecology of marine, coastal and lake ecosystems, including detailed research on acid precipitation and lake acidification (*Flower & Battarbee, 1983*; *Battarbee et al., 1988*; *Renberg, 1990*), lake salinity (*Fritz et al., 1991*; *Gasse, Juggins & Khelifa, 1995*), total nitrogen and phosphorus (*Bennion, Juggins & Anderson, 1996*), coastal environmental change (*Lewis et al., 2013*) and ecotoxicology (*Pandey et al., 2017*). Diatoms have also been used extensively in the assessment of environmental conditions of riverine and lentic ecosystems (*Stevenson, Pan & Van Dam, 2010*; *Bennion et al., 2014*). Research in shallow streams usually involves benthic diatom assemblages, whereas planktic diatoms are predominantly studied in research centred on larger rivers. Studies assessing downstream effects of river impoundment using diatom datasets are relatively scarce (*Growns & Growns, 2001*). Planktic diatom assemblages have often been studied within reservoirs (*Tolotti, Boscaini & Salmaso, 2010*; *Fornarelli & Antenucci, 2011*), with some reporting changes downstream of dams on regulated rivers (e.g., *Tornés et al., 2014*). Studies of benthic habitats, including epilithic, epiphytic and epipelic diatom assemblages (growing on rocks, aquatic plants and sediments, respectively) within impounded rivers are more common, including research on assemblage structure (e.g., *Blinn, Truitt & Pickart, 1989*; *Growns, 1999*; *Gallo et al., 2015*) and biomass (*Uehlinger, Kawecka & Robinson, 2003*; *Nichols et al., 2006*) downstream of hydropower (*Peterson, 1986*; *Wu et al., 2009*) and water supply structures (*Boix et al., 2010*). Comparisons of pre- and post-impoundment diatom assemblages are rare in the scientific literature due to a general absence of pre-impoundment monitoring data (but see *Cibils Martina, Principe & Gari, 2013*; *Algarte et al., 2016*). As a result, most studies have used non-regulated control sites (upstream or on unregulated tributaries) that are assumed to represent reference conditions at downstream sites.

Most research on the effects of river impoundment on benthic diatom assemblages has been undertaken at local scales, focusing on single or several reservoirs and involving short time periods, with few studies deploying large-scale or long-term datasets (e.g., *Growns*

& *Growns, 2001*; *Marzin et al., 2012*; *Dahm et al., 2013*). Although diatoms are often used to investigate water quality and nutrient pressures, including total phosphorus in rivers (e.g., *Kelly & Whitton, 1995*; *Szczepocka, Kruk & Rakowska, 2015*), diatoms are known to be sensitive to other pressures including river discharge (e.g., *Snell et al., 2014*). A number of river impoundment studies using diatom datasets have established direct links between diatom assemblage responses and river flow regime modification (e.g., *Growns, 1999*; *Wu et al., 2010*). Most water supply reservoirs release managed discharges (termed 'compensation flows' in a UK context— *Gustard, 1989*; *Acreman & Dunbar, 2004*) that bear little resemblance to the natural flow regime, and that typically reduce high and median flows and increase low flows (*Higgs & Petts, 1988*; *McManamay, Orth & Dolloff, 2012*; *Stewardson et al., 2017*). The effect of these flow changes downstream of water supply reservoirs on diatom assemblages remains poorly understood, and transferable flow–ecology relationships are difficult to formulate at larger spatial scales (*Poff & Zimmerman, 2010*; *Webb et al., 2013*).

In this paper, we compare benthic diatom assemblages at monitoring sites downstream of multiple water supply reservoirs subject to fixed-flow releases with assemblages at paired unregulated control sites. We hypothesised that diatom assemblages at downstream sites would differ from those at control sites, and aimed to identify consistent patterns in downstream diatom responses beyond the scale of individual reservoirs. In addition, we hypothesised that ecological guilds based on growth forms would respond differently to river impoundment, which may help elucidate the role of abiotic factors in shaping diatom assemblages. To investigate this, we used a large-scale national-scale biomonitoring dataset (covering 2013–2016) associated with 77 reservoirs in England. The study addressed the following research questions: (1) Do consistent differences exist between diatom assemblages at sampling sites downstream of water supply reservoirs and unregulated control sites at the regional and national scale? (2) Can diatom assemblage structure and ecological guilds be used to identify environmental gradients (e.g., flow, water quality) associated with sites downstream of impoundments and unregulated control sites?

## METHODS

### Study area

We used data from the WRPN monitoring network, which is short for Water Resource Permitting and Planning Network. The network was established by the Environment Agency of England (EA), the statutory environmental regulatory agency in England, UK, at a subset of compensation release reservoirs, predominantly located in upland areas of England, to provide monitoring data to develop the understanding of the ecological impacts of downstream flow alteration. The most important feature of WRPN is the pairing of biomonitoring sites downstream of water supply reservoirs (referred to as 'downstream sites' hereafter) and control sites. Only monitoring sites with sufficiently good water quality were selected by the EA in order to minimise possible confounding effects of background water quality issues on biomonitoring results (i.e., no major pesticide, acidification or pollution incidents at or near the sites prior to or during the study period). As no pre-impoundment biomonitoring took place historically, no before–after comparison could be

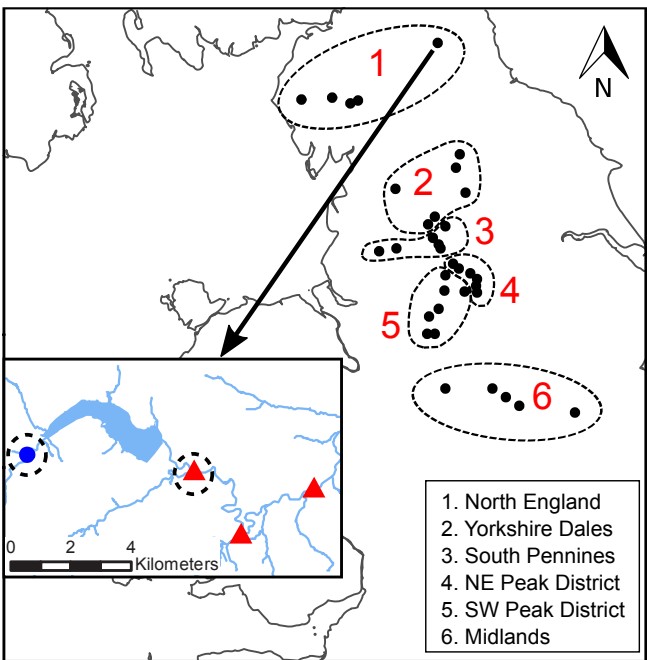

**Figure 1 Location of the WRPN reservoir clusters.** The dash-lined circles indicate the six regions. IN-SET: close-up of the Derwent reservoir in Northumberland with one control site (blue dot) and three downstream sites (red triangles). The dashed circles indicate the sample sites that were selected from this cluster.

established. Hence, control sites located either on river reaches upstream of the reservoirs or on adjacent unregulated tributaries were selected to reflect the conditions that would occur at the downstream sites without the presence of the reservoir (see also *Krajenbrink et al., 2019*). We selected 34 clusters comprising individual or serial impoundments from the total WRPN network (Fig. 1). In instances where a cluster comprised multiple downstream and control sites, one downstream and one control site closest to the impoundment were selected, resulting in a total of 68 sample sites (34 downstream–control site pairs) associated with 77 reservoirs (one to six reservoirs per cluster). In total, 25 control sites were located upstream of the reservoirs and nine sites were located on nearby tributaries.

## Diatom sampling

All sites were sampled biannually during spring (predominantly April–May, with a few samples collected in March) and autumn (September–November) between 2013 and 2016. This period had average to above-average annual rainfall, with no prolonged dry periods (*Met Office, 2019*). A maximum of eight epilithic diatom samples were collected per site. For sites that were introduced or replaced in 2015 after a network revision ($n = 18$; mainly control sites), samples were only available for 2015 and 2016 (maximum of four samples). Diatom sampling followed EA protocols (*WFD-UKTAG, 2014*) which conform to European standards for routine diatom sampling and preparation in rivers and lakes (*CEN, 2014b*). At each site, a minimum of five cobbles or small boulders were selected. Diatoms were removed

from the cobbles using a small brush, preserved in the field using Lugol's iodine solution, and returned to the laboratory for processing and identification. In the laboratory, diatom frustules were cleaned using a combination of acids and were prepared on slides. Samples were identified using a compound light microscope at 1,000× magnification by counting at least 300 valves of benthic diatom taxa, comprising at least 200 valves of non-dominant taxa. This process was compliant with European standards on the identification of benthic diatoms from rivers and lakes (*CEN, 2014a*). For a randomly selected number of samples, slide preparation and diatom identifications were independently verified in an external audit procedure following EA quality assurance protocols. Diatom valves were identified to species level where possible, with taxa that formed <2% of the sample typically being resolved to genus level.

A total of 477 samples were available for analysis (244 samples from downstream sites, 233 control site samples). Taxa with outdated nomenclature were merged with currently valid taxonomic entries to ensure a consistent taxonomy across all samples. In addition, a number of rare or planktic genera were resolved to genus level, and genus entries that overlapped with species entries within the same genus were removed. The final diatom dataset comprised 379 taxa including 18 genus entries.

## DATA ANALYSIS

Diatom assemblages at sites downstream of impoundments and at control sites were analysed at different spatial scales. At the national scale, all 34 downstream–control site pairs (504 samples) were analysed in combination. At the regional scale, the 34 pairs were divided into six regions (see Fig. 1), each region comprising five to seven pairs, following the principles of *Krajenbrink et al. (2019)*. Prior to analysis, all diatom samples were transformed into relative counts. The final diatom dataset included 23 planktic taxa (following the checklist in *Rimet & Bouchez, 2012b*), which were excluded from the analysis steps unless explicitly mentioned, in order to focus on benthic taxa.

### Assemblage structure

Differences in benthic diatom assemblage structure were tested against three spatial (*SiteType*, *Region* and *DCpair* or 'Downstream–Control site pair') and two temporal variables (Season and Year). A three-step approach similar to *Krajenbrink et al. (2019)* was used, but is summarised here for clarity: (1) The influence of *SiteType* (downstream or control site) on diatom assemblages was examined by analysing spring and autumn samples separately; (2) The effect of impoundment on seasonal patterns within diatom assemblage structure was examined by dividing samples into 'Downstream' and 'Control' site groups and testing both groups in association with *Season* (spring or autumn); (3) The influence of other spatio-temporal variables *Region*, *Year* and *DCpair* was tested by analysing both seasons and site types individually, thus dividing samples into four groups.

Differences in multivariate assemblage structure between levels of individual variables were tested with a non-parametric Permutational Multivariate Analysis of Variance (PERMANOVA; see *Anderson, 2001*) using the `adonis` function from the R package **vegan** (*Oksanen et al., 2017*), which partitions the total statistical variation (sums of

squares) for the different sources of variation. Using this approach enabled the amount of total statistical variation explained by the spatio-temporal variables and the unexplained variation to be determined (see also *Krajenbrink et al., 2019*). The partitioned statistical variation, expressed as partial *R*-squared values or $R^2$-values by the `adonis` function, will be presented throughout the text and alongside the standard pseudo-*F* and *p*-value output in the relevant tables. The significance of the PERMANOVA partitions were tested using 999 permutations. Multivariate patterns in assemblage structure were visualised using Non-metric Multidimensional Scaling (NMDS) ordination, using the `metaMDS` function in **vegan**. Both PERMANOVA and NMDS were applied using Bray–Curtis dissimilarity matrices (see also *Krajenbrink et al., 2019*).

## Diatom indices, guilds and indicator taxa

A number of univariate indices based on diatom assemblage composition were derived. We determined the total taxonomic richness of benthic diatoms (*Ntaxa*) for each sample. Taxonomic richness was investigated in more detail by segregating all benthic taxa into three diatom ecological guilds based on growth morphologies and their ability to tolerate nutrient limitation and physical disturbance, i.e., low-profile, high-profile, and motile guilds (*Passy, 2007*; *Rimet & Bouchez, 2012b*). The low-profile guild comprises diatom taxa of short stature that are predominantly attached to substrate. Diatom taxa belonging to the high-profile guild are larger or tend to form long colonies or stalks. The motile guild comprises comparatively fast-moving species. It is hypothesised that the specific guilds would respond differently to flow velocity conditions and changes therein, with low-profile taxa expected to occur more frequently in high-flow conditions (high disturbance) and high-profile taxa favouring tranquil flow (low disturbance) conditions (*Passy, 2007*). For every sample, the absolute number as well as the percentage of taxa belonging to each guild (relative richness –*%taxa_low; %taxa_high; %taxa_motile*) were calculated. The diatom checklist provided by *Rimet & Bouchez (2012b)* was used to assign each taxon to a guild. Indicator taxa for each region and season were derived for both site types using the `multipatt` function from the R package **indicspecies** (*De Caceres & Jansen, 2016*) to determine the diatom taxa that were characteristic of downstream and control sites.

In addition to the richness and guild indices, Trophic Diatom Index (TDI) scores were derived. The TDI methodology was initially developed to investigate the trophic status of rivers (*Kelly & Whitton, 1995*) and has been used to assess phytobenthic ecosystems in UK waterbodies for the WFD (*Kelly et al., 2008*; *WFD-UKTAG, 2014*). TDI scores (version TDI4) were calculated by the EA following guidelines in *WFD-UKTAG (2014)*. Finally, the relative abundance of planktic diatoms per sample (*%planktic*) was calculated, expressed as percentages of the total number of benthic and planktic diatoms cells per sample. Differences between diatom indices at downstream and control sites were tested by means of a non-parametric one-way Analysis of Variance (Kruskal–Wallis), for which spring and autumn samples were analysed separately (see also *Krajenbrink et al., 2019*).

## Water quality variables

Measurements of water quality were extracted from the water quality monitoring network maintained by the EA (one measurement per sample season). Six variables were extracted

**Table 1  Results from PERMANOVA testing the significance of SiteType on assemblage structure for separate spring and autumn samples (Step 1).**

| Region | Season | pseudo-$F$ | $R^2$ | $p$ |
|---|---|---|---|---|
| National scale | Spring | 8.33 | 0.03 | **0.001**[***] |
| | Autumn | 5.00 | 0.02 | **0.001**[***] |
| North England | Spring | 1.62 | 0.05 | 0.101 |
| | Autumn | 1.88 | 0.06 | **0.041**[*] |
| Yorkshire Dales | Spring | 1.92 | 0.05 | **0.021**[*] |
| | Autumn | 1.62 | 0.04 | 0.098 |
| South Pennines | Spring | 4.50 | 0.10 | **0.001**[***] |
| | Autumn | 4.46 | 0.11 | **0.002**[**] |
| NE Peak District | Spring | 8.98 | 0.16 | **0.001**[***] |
| | Autumn | 8.02 | 0.14 | **0.001**[***] |
| SW Peak District | Spring | 3.24 | 0.09 | **0.002**[**] |
| | Autumn | 1.32 | 0.04 | 0.186 |
| Midlands | Spring | 4.06 | 0.11 | **0.002**[**] |
| | Autumn | 2.58 | 0.07 | **0.010**[*] |

**Notes.**

Significant $p$-values are in bold font.

[***] $p \leq 0.001$.

[**] $p \leq 0.01$.

[*] $p \leq 0.05$.

for detailed analysis: conductivity ($\mu Scm^{-1}$); un-ionised ammonia ($NH_3$—$mgL^{-1}$); total oxidised nitrogen (TON—$mgL^{-1}$); orthophosphate (OP—$mgL^{-1}$); dissolved oxygen ($O_2$–$mgL^{-1}$); and pH (-). Given that water quality monitoring was undertaken independently from the diatom sampling, not every diatom sample could be paired with a water quality sample (80–88% coverage, depending on water quality variable, with a maximum offset of 60 days). Differences in water quality values between different site types (downstream or control sites) were tested using Kruskal–Wallis, analysing spring and autumn samples separately. In addition, correlations between water quality variables and diatom indices were calculated. Values for conductivity, $NH_3$, TON and OP were log-transformed prior to analysis for plotting purposes.

# RESULTS

## Assemblage structure

We found that benthic diatom assemblage structure at downstream sites was significantly different from control sites during both spring and autumn (Step 1 of analysis) at the national scale. Variable *SiteType* explained 2–3% of the statistical variation (see Table 1 for PERMANOVA $R^2$ values). The clusters of downstream and control site samples in the NMDS plots largely overlapped, although the density and distribution of both point clusters was different (autumn samples in Fig. 2—see Fig. S1 for spring samples). A similar pattern was observed at the regional scale, with diatom assemblage structure at downstream sites being significantly different to control sites in most regions (Table 1). Higher $R^2$ values for *SiteType* were recorded at the regional scale than at the national scale (4–16%), with the

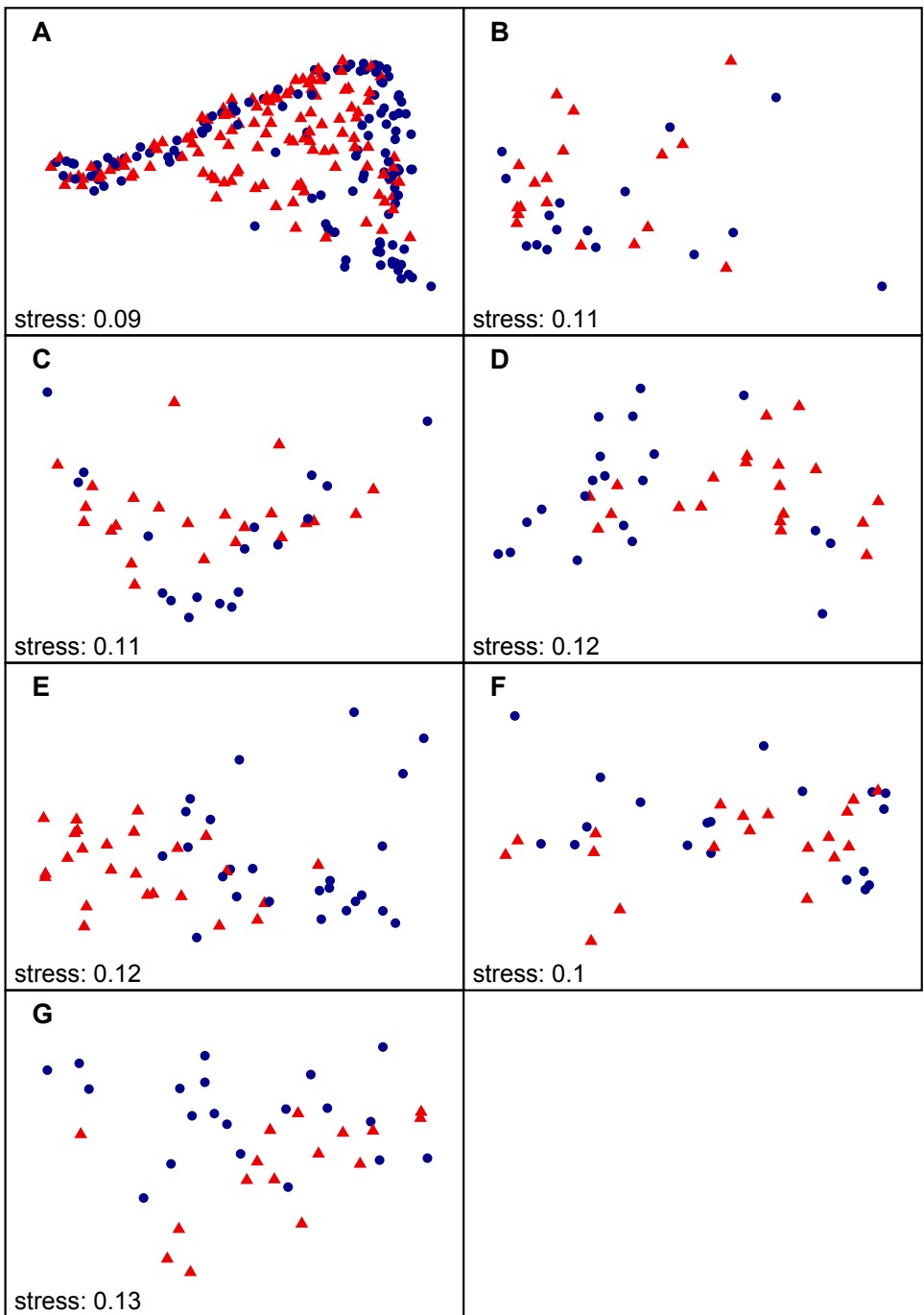

**Figure 2** **NMDS ordination results including all autumn samples on national (A) and region scale (B–H), labelled according to SiteType (Step 1).** (A) National scale; (B) North England; (C) Yorkshire Dales; (D) South Pennines; (E) North East peak District; (F) South West Peak District; (G) Midlands. Blue dots = control sites, red triangles = downstream sites.

**Table 2** Results from PERMANOVA testing the significance of Season on assemblage structure for separate control (C) and downstream site (D) samples (Step 2).

| Region | Site type | pseudo-$F$ | $R^2$ | $p$ |
|---|---|---|---|---|
| National scale | C | 6.57 | 0.03 | **0.001**[***] |
| | D | 12.39 | 0.05 | **0.001**[***] |
| North England | C | 1.24 | 0.04 | 0.254 |
| | D | 3.18 | 0.09 | **0.002**[**] |
| Yorkshire Dales | C | 1.84 | 0.05 | **0.048**[*] |
| | D | 4.99 | 0.10 | **0.001**[***] |
| South Pennines | C | 3.45 | 0.08 | **0.002**[**] |
| | D | 3.57 | 0.09 | **0.001**[***] |
| NE Peak District | C | 4.14 | 0.08 | **0.001**[***] |
| | D | 4.32 | 0.08 | **0.001**[***] |
| SW Peak District | C | 2.57 | 0.07 | **0.013**[*] |
| | D | 3.17 | 0.08 | **0.001**[***] |
| Midlands | C | 9.08 | 0.20 | **0.001**[***] |
| | D | 6.40 | 0.17 | **0.001**[***] |

**Notes.**
Significant $p$-values are in bold font.
[***] $p \leq 0.001$.
[**] $p \leq 0.01$.
[*] $p \leq 0.05$.

highest values observed in the North East Peak District region, and clusters of downstream and control site samples in the NMDS plots were separate for most regions (autumn samples in Fig. 2; see Fig. S1 for spring samples).

In step 2, we observed significant differences in diatom assemblage structure between spring and autumn samples at the national scale. Variable *Season* explained a slightly greater proportion of the total statistical variation ($R^2$) for downstream (5%) than for control sites (3%) (Table 2). The effect of *Season* was also significant at the regional scale for all regions except for control sites in North England (Table 2). The highest $R^2$ values were typically observed at downstream sites (8–17%), although the highest value was recorded for control sites in the Midlands region (20%).

Significant differences in assemblage structure were observed between the various regions (Step 3), with $R^2$ values for *Region* being typically lower for downstream site samples (17–22%) than for control sites (25–30%). For both site types, $R^2$ values were higher for autumn samples than for spring samples (Tables S1–S2). Samples from the individual regions formed largely separate clusters in the ordination plots (Fig. 3). The variable Year was not significant at either the national or regional scale. We also found that variable *DCpair* was significant, with high $R^2$ values at the national scale (54–68%). $R^2$ values were higher at control sites and higher for autumn samples at both downstream and control sites. *DCpair* was also statistically significant at the regional scale, explaining a large proportion of the statistical variation, although this was lower than at the national scale. Marked variations in $R^2$ values were observed between regions. For control sites, the
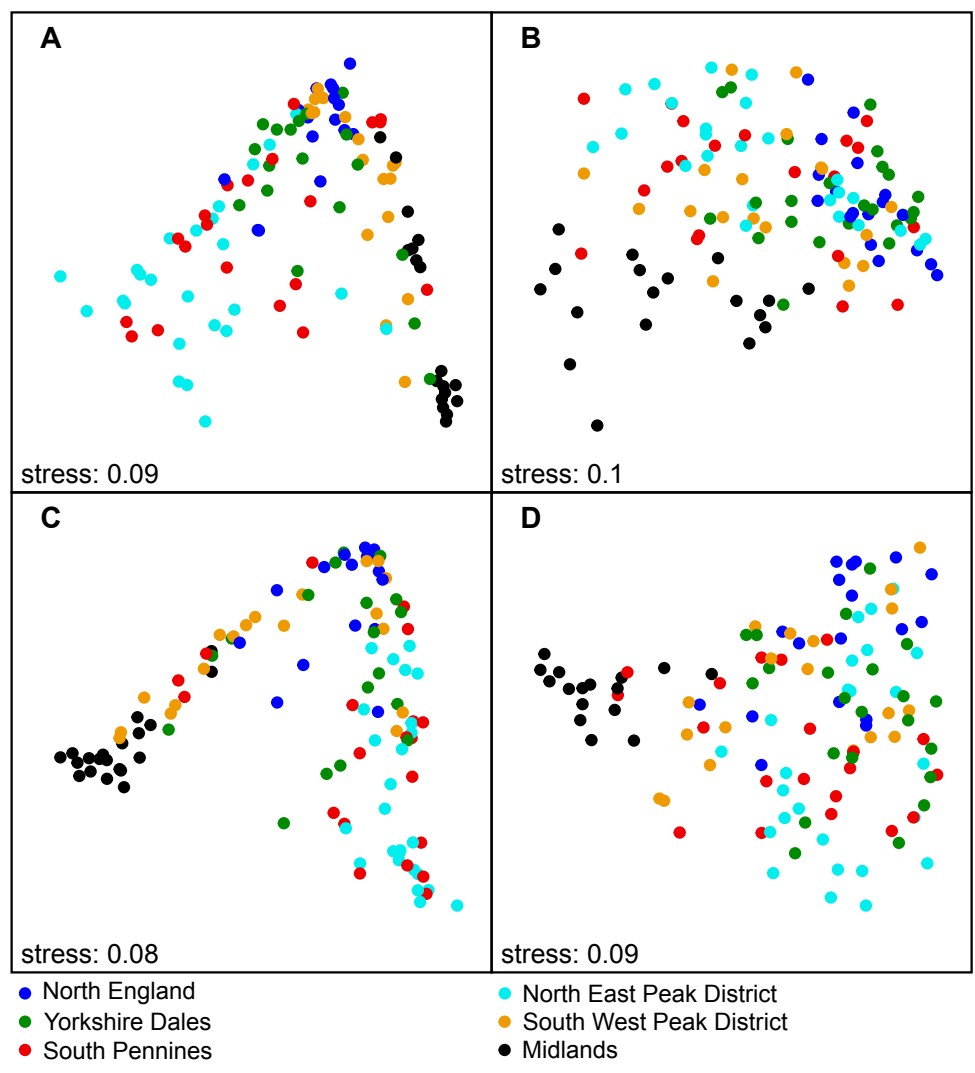

**Figure 3** National-scale NMDS ordination results involving sample subsets based on SiteType and Season, labelled according to Region (Step 3). (A) Control sites—spring; (B) Downstream sites—spring; (C) Control sites—autumn; (D) Downstream sites –autumn.

$R^2$ was generally higher for autumn samples, with a more variable pattern at downstream sites.

## Taxonomic richness and ecological guilds

We observed significant differences in diatom taxonomic richness (Ntaxa) between downstream sites and control sites at the national scale (Table 3), with on average significantly higher values at downstream sites (27–33 taxa) than at control sites (22–24 taxa). In addition, *Ntaxa* was typically higher during autumn than spring, especially at downstream sites. At the regional scale, we found that *Ntaxa* was significantly diffferent at downstream sites compared to control sites during both spring and autumn for all regions except Midlands, with values on average higher at downstream than at control sites (autumn

samples in Fig. 4; spring samples in Fig. S2). In addition, *Ntaxa* at downstream sites was higher during autumn in most instances, while no clear seasonal pattern was detected at control sites. When ecological guilds were considered, it was found that taxonomic richness for all three guilds was typically higher at downstream sites, but most notably for the motile guild (Table S3).

The proportion of high-profile taxa (*%taxa_high*) was greater than the other two ecological guilds, on average just below 50%, at the national scale as well as for most regions (autumn samples in Fig. 5; spring samples in Fig. S3). Values for *%taxa_low* and *%taxa_motile* were typically around 30% (Table 4). Values of *%taxa_high* and *%taxa_low* were typically lower at downstream sites than at control sites at both national and regional scales, but differences were limited. Lower downstream values were significant at the national scale and in two to three regions for *%taxa_high* and one to two regions for *%taxa_low* (Kruskal–Wallis). In contrast, values for *%taxa_motile* were generally higher at downstream sites than control sites (autumn samples in Fig. 5; spring samples in Fig. S3; Table 4), with the differences between site types being clearer than for the high- and low-profile guilds. Higher downstream values were significant at the national scale and in three regions during spring and four regions during autumn. Results for the Midlands region differed from the other regions, with *%taxa_high* and *%taxa_low* being slightly higher at downstream sites (significant only for *%taxa_low* in autumn), while values for *%taxa_motile* were typically higher at control sites; significantly so for autumn samples (Table 4). Values for *%taxa_high* were typically higher during spring than autumn at both site types, while *%taxa_low* and *%taxa_motile* were higher during autumn, both at the national scale and in most regions and site types.

*%taxa_high* displayed strong negative correlations with *%taxa_motile* and TDI for both seasons and a moderate correlation with Ntaxa in autumn, while *%taxa_motile* was positively correlated with *Ntaxa* and TDI (Table S4). Values for *%taxa_low* did not display any clear correlation with any other diatom indices.

### Indicator taxa

From the 356 benthic diatom taxa in the dataset, 75 taxa were identified as indicators of site type (downstream or control) in one or more regions or seasons. The full indicator taxa list, as well as the site type for which they are indicators in the individual region and season, can be found in Appendix SB. More indicator taxa were identified for downstream sites (55) than for control sites (25), with the largest number of indicator taxa recorded at downstream sites in the South Pennines and North East Peak District during autumn. Important indicator taxa at downstream sites, found for multiple regions and seasons, included *Diatoma problematica*, *Encyonema minutum*, *Melosira varians*, *Nitzschia dissipata* and *Nitzschia fonticola*. At control sites, important indicator taxa included *Eunotia exigua* and *Pinnularia subcapitata*. Almost all taxa were indicators for just one site type, with just five taxa found as indicators for both control and downstream sites in different regions. Four taxa (*Navicula cryptocephala*, *Nitzschia linearis*, *Nitzschia sociabilis* and *Surirella brebissonii*) were indicators of control sites in the Midlands region, but indicators of downstream sites in some other regions. At downstream sites, more indicator taxa were recorded

Krajenbrink et al. (2019), *PeerJ*, DOI 10.7717/peerj.8092

**Table 3** Mean values (± sd) of taxonomic richness (Ntaxa), TDI and relative abundance of planktic diatoms (%planktic) for control (C) and downstream (D) sites, as well as results from Kruskal–Wallis ($\chi^2$ and $p$) testing the significance of variable SiteType on these variables for separate spring and autumn samples.

| Region | Season | Ntaxa | | | | TDI | | | | %planktic | | | |
|---|---|---|---|---|---|---|---|---|---|---|---|---|---|
| | | C | D | $\chi^2$ | $p$ | C | D | $\chi^2$ | $p$ | C | D | $\chi^2$ | $p$ |
| National scale | Spring | 22 ± 8 | 27 ± 7 | 29.6 | **0.001**[***] | 35 ± 18 | 38 ± 16 | 4.1 | **0.042**[*] | 4 ± 8 | 5 ± 6 | 7.6 | **0.006**[**] |
| | Autumn | 24 ± 9 | 33 ± 8 | 51.3 | **0.001**[***] | 34 ± 23 | 42 ± 19 | 10.9 | **0.001**[***] | 3 ± 6 | 8 ± 12 | 33.8 | **0.001**[***] |
| North England | Spring | 20 ± 5 | 27 ± 6 | 10.2 | **0.001**[***] | 21 ± 7 | 22 ± 7 | 0.6 | 0.448 | 2 ± 3 | 8 ± 10 | 2.9 | 0.090 |
| | Autumn | 19 ± 7 | 30 ± 8 | 11.4 | **0.001**[***] | 24 ± 15 | 31 ± 13 | 2.7 | 0.098 | 5 ± 4 | 12 ± 12 | 2.3 | 0.127 |
| Yorkshire Dales | Spring | 21 ± 7 | 26 ± 4 | 5.0 | **0.026**[**] | 29 ± 18 | 30 ± 10 | 1.1 | 0.293 | 6 ± 10 | 4 ± 5 | 0.0 | 0.862 |
| | Autumn | 21 ± 8 | 33 ± 6 | 15.2 | **0.001**[***] | 29 ± 16 | 34 ± 14 | 1.5 | 0.217 | 5 ± 11 | 4 ± 4 | 2.1 | 0.148 |
| South Pennines | Spring | 18 ± 8 | 26 ± 9 | 10.2 | **0.001**[***] | 29 ± 12 | 39 ± 13 | 5.8 | **0.016**[**] | 6 ± 11 | 6 ± 8 | 0.7 | 0.414 |
| | Autumn | 23 ± 10 | 37 ± 9 | 14.5 | **0.001**[***] | 23 ± 14 | 42 ± 17 | 9.2 | **0.002**[**] | 4 ± 10 | 8 ± 13 | 5.8 | **0.016**[*] |
| NE Peak District | Spring | 22 ± 7 | 26 ± 5 | 5.3 | **0.022**[*] | 28 ± 10 | 36 ± 11 | 4.2 | **0.041**[*] | 2 ± 4 | 4 ± 3 | 7.3 | **0.007**[**] |
| | Autumn | 21 ± 7 | 33 ± 8 | 20.7 | **0.001**[***] | 16 ± 9 | 35 ± 10 | 24.2 | **0.001**[***] | 1 ± 2 | 10 ± 12 | 18.2 | **0.001**[***] |
| SW Peak District | Spring | 21 ± 6 | 28 ± 6 | 9.6 | **0.002**[**] | 38 ± 14 | 40 ± 12 | 0.4 | 0.543 | 3 ± 6 | 5 ± 6 | 3.7 | 0.055 |
| | Autumn | 23 ± 7 | 33 ± 10 | 8.6 | **0.003**[**] | 44 ± 19 | 44 ± 18 | 0.0 | 0.895 | 2 ± 3 | 11 ± 19 | 9.1 | **0.003**[**] |
| Midlands | Spring | 30 ± 6 | 32 ± 9 | 0.7 | 0.409 | 63 ± 12 | 63 ± 10 | 0.0 | 0.950 | 6 ± 10 | 3 ± 4 | 0.1 | 0.733 |
| | Autumn | 35 ± 6 | 31 ± 8 | 2.3 | 0.129 | 73 ± 5 | 72 ± 10 | 0.0 | 0.824 | 1 ± 2 | 2 ± 4 | 0.7 | 0.406 |

**Notes.**

Significant *p*-values are in bold font.

[***] $p \leq 0.001$.

[**] $p \leq 0.01$.

[*] $p \leq 0.05$.

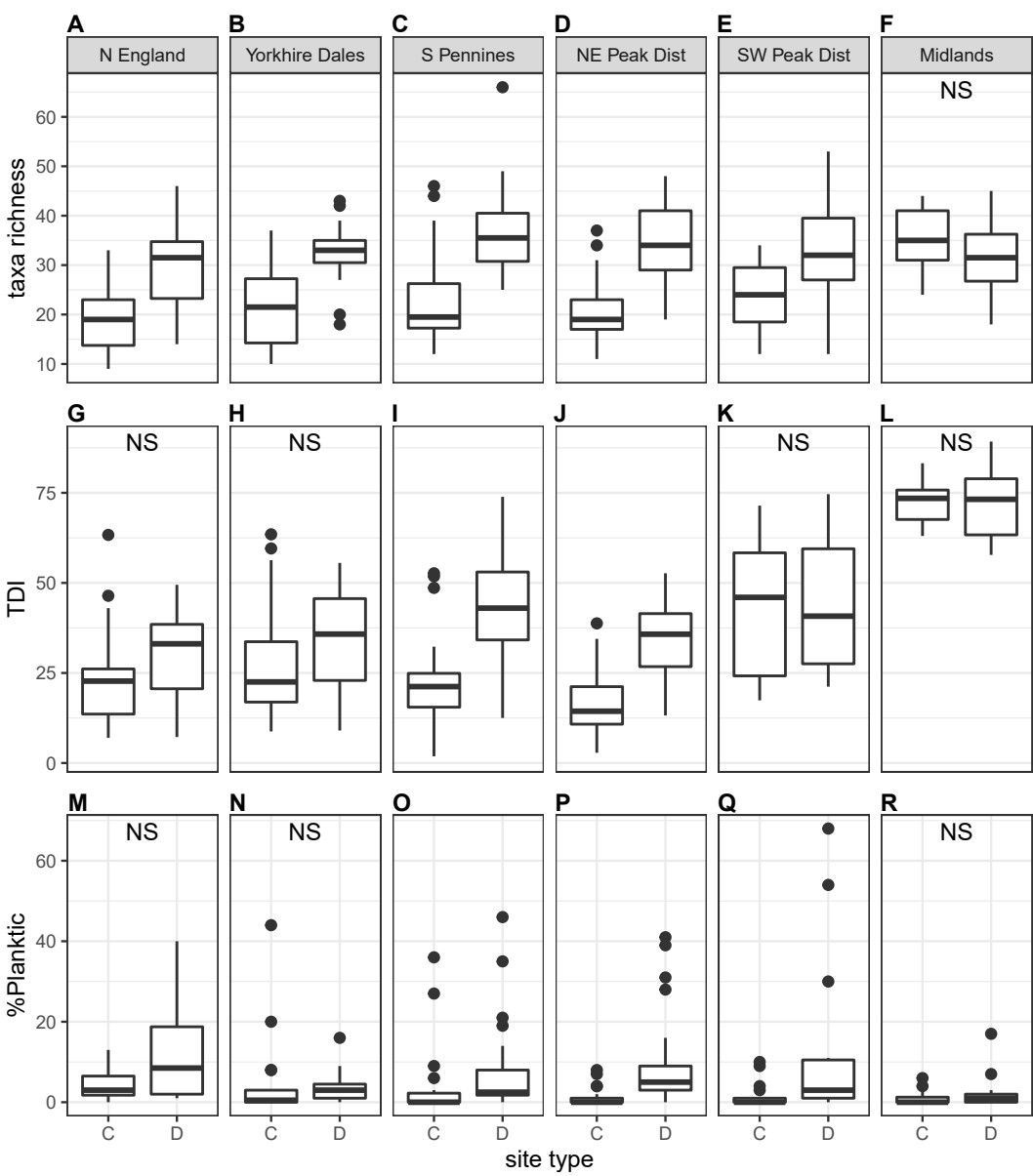

**Figure 4** Regional-scale values for taxa richness (A–F), TDI (G–L) and %planktic (M–R) calculated on autumn samples, per site type (C = control, D = downstream sites). Non-significant Kruskal–Wallis results are indicated with 'NS'.

during autumn than spring for all regions except Midlands, while no clear pattern was observed for control sites. Indicator taxa were typically ordered by diatom genus. Species within the genera *Eunotia*, *Frustulia* and *Pinnularia* were typical indicators of control sites. Genera of indicator species at downstream sites included *Cocconeis*, *Diatoma*, *Navicula*, *Nitzschia* and *Planothidium*. The genus *Gomphonema* comprised both indicators of control sites and downstream sites. No clear relationship was observed between indicator taxa and ecological guilds. Taxa in the low-profile and to a lesser extent motile guild were

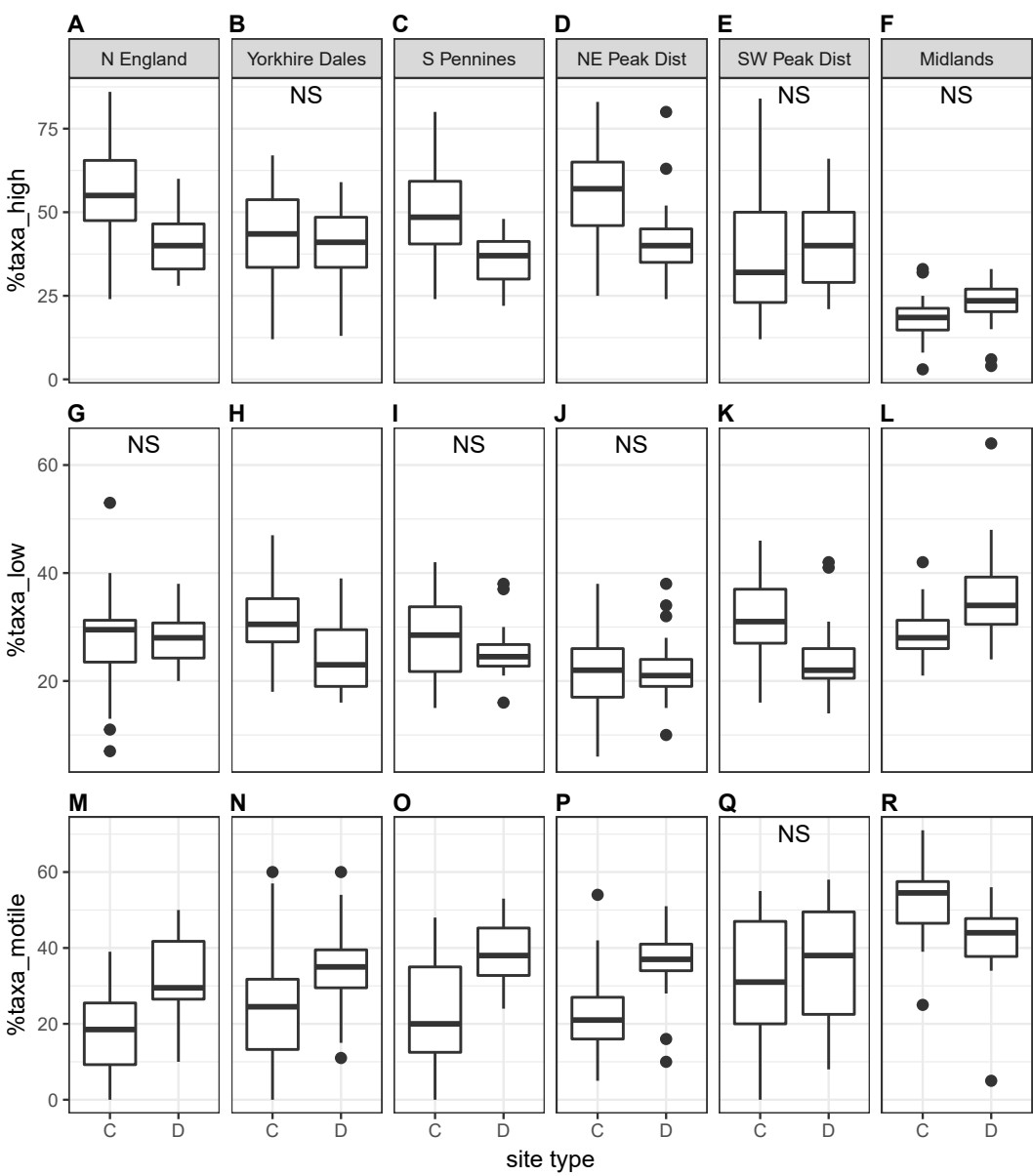

**Figure 5** Regional-scale values for percentages of high-profile (A–F), low-profile (G–L) and motile taxa (M–R) calculated on autumn samples, per site type (C = control, D = downstream sites). Non-significant Kruskal–Wallis results are indicated with 'NS'.

predominantly indicators of downstream sites, while the high-profile guild comprised both indicators of downstream and control sites.

## TDI and planktic diatoms

Trophic Diatom Index (TDI) values were typically higher at downstream sites than at control sites, both at the national scale and in most regions (autumn samples in Fig. 4; see Fig. S2 for spring samples). The difference was statistically significant at the national scale and in the South Pennines and North East Peak District regions (Table 3). Values

Krajenbrink et al. (2019), *PeerJ*, DOI 10.7717/peerj.8092

**Table 4** Mean values (± sd) of the percentages of ecological guilds for control (C) and downstream (D) sites, as well as results from Kruskal–Wallis ($\chi^2$ and $p$) testing the significance of variable SiteType on those percentages for separate spring and autumn samples.

| Region | Season | %taxa_high | | | | %taxa_low | | | | %taxa_motile | | | |
|---|---|---|---|---|---|---|---|---|---|---|---|---|---|
| | | C | D | $\chi^2$ | $p$ | C | D | $\chi^2$ | $p$ | C | D | $\chi^2$ | $p$ |
| National scale | Spring | 45 ± 18 | 42 ± 12 | 4.0 | **0.045**[*] | 27 ± 8 | 25 ± 7 | 5.5 | **0.019**[*] | 27 ± 17 | 33 ± 12 | 12.2 | **0.001**[***] |
| | Autumn | 43 ± 19 | 38 ± 12 | 6.4 | **0.012**[*] | 28 ± 8 | 26 ± 8 | 5.3 | **0.021**[*] | 29 ± 18 | 36 ± 12 | 14.4 | **0.001**[***] |
| North England | Spring | 59 ± 11 | 50 ± 9 | 3.2 | 0.074 | 31 ± 8 | 29 ± 7 | 0.9 | 0.351 | 11 ± 6 | 22 ± 8 | 14.7 | **0.001**[***] |
| | Autumn | 54 ± 15 | 40 ± 9 | 8.7 | **0.003**[*] | 27 ± 12 | 28 ± 5 | 0.1 | 0.769 | 19 ± 12 | 32 ± 11 | 7.7 | **0.005**[**] |
| Yorkshire Dales | Spring | 45 ± 11 | 49 ± 11 | 1.2 | 0.269 | 32 ± 5 | 23 ± 6 | 15.5 | **0.001**[***] | 23 ± 13 | 28 ± 10 | 1.1 | 0.299 |
| | Autumn | 43 ± 16 | 40 ± 12 | 0.6 | 0.438 | 31 ± 7 | 25 ± 6 | 7.7 | **0.006**[**] | 26 ± 18 | 35 ± 12 | 4.9 | **0.027**[*] |
| South Pennines | Spring | 55 ± 16 | 42 ± 10 | 8.2 | **0.004**[**] | 24 ± 8 | 24 ± 7 | 0.0 | 0.946 | 21 ± 15 | 34 ± 10 | 10.2 | **0.001**[***] |
| | Autumn | 49 ± 14 | 36 ± 7 | 8.7 | **0.003**[**] | 28 ± 8 | 26 ± 5 | 0.9 | 0.343 | 23 ± 15 | 38 ± 9 | 10.4 | **0.001**[***] |
| NE Peak District | Spring | 53 ± 11 | 44 ± 7 | 10.8 | **0.001**[***] | 23 ± 7 | 21 ± 7 | 1.2 | 0.281 | 24 ± 10 | 35 ± 8 | 15.7 | **0.001**[***] |
| | Autumn | 55 ± 13 | 42 ± 12 | 14.0 | **0.001**[***] | 22 ± 7 | 22 ± 6 | 0.0 | 0.930 | 22 ± 11 | 36 ± 9 | 16.1 | **0.001**[***] |
| SW Peak District | Spring | 37 ± 18 | 40 ± 12 | 0.8 | 0.361 | 28 ± 8 | 24 ± 6 | 2.0 | 0.152 | 35 ± 19 | 36 ± 13 | 0.0 | 0.927 |
| | Autumn | 38 ± 20 | 41 ± 13 | 0.9 | 0.335 | 32 ± 8 | 24 ± 7 | 9.9 | **0.002**[**] | 30 ± 18 | 35 ± 16 | 0.7 | 0.389 |
| Midlands | Spring | 21 ± 10 | 25 ± 9 | 1.3 | 0.254 | 29 ± 8 | 30 ± 7 | 0.3 | 0.601 | 50 ± 9 | 45 ± 7 | 2.1 | 0.149 |
| | Autumn | 19 ± 8 | 22 ± 8 | 3.2 | 0.072 | 29 ± 5 | 36 ± 10 | 7.1 | **0.008**[**] | 52 ± 10 | 42 ± 12 | 7.7 | **0.006**[**] |

**Notes.**
Significant $p$-values are in bold font.
[***] $p \leq 0.001$.
[**] $p \leq 0.01$.
[*] $p \leq 0.05$.

were generally higher for autumn samples, although differences were small and the effect was marginal at control sites. The difference between control and downstream sites was stronger in autumn. Scores were markedly higher for the Midlands than for all other regions. In addition to a positive correlation with *%taxa_motile* and a negative correlation with *%taxa_high* (see also section above), a moderate correlation with *Ntaxa* was observed during autumn (Table S4).

The proportion of planktic diatoms (*%planktic*) in samples was generally low (autumn samples in Fig. 4; see Fig. S2 for spring samples), with the lowest values recorded at control sites (Table 3). This difference was significant at the national scale, with *%planktic* at control sites being higher in spring than in autumn. The opposite trend was recorded at downstream sites, where autumn samples were generally characterised by higher values for *%planktic*. A similar pattern was observed at the regional scale with average values typically higher at downstream sites than control sites. The highest values were recorded for autumn samples at downstream sites, except for the Midlands region. The difference was significant for only one region during spring and three regions during autumn (Table 3). No clear correlations were observed between *%planktic* and any other diatom indices (Table S4).

## Water quality variables

Overall, no consistent difference was observed among water quality variables between downstream sites and control sites. Values for conductivity, total oxidised nitrogen (TON) and ammonia ($NH_3$) were typically higher at downstream sites in most regions, although not statistically significant in most instances (Figs. S4–S9; Table S5 ). Values for dissolved oxygen ($O_2$) were lower at downstream sites during autumn (significant in two out of six regions), but no clear pattern was detected for spring samples. Values for conductivity, TON, $NH_3$ and orthophosphate (OP) were markedly higher in the Midlands region and displayed a greater spread. No clear pattern was detected for pH values.

From the diatom response variables examined, TDI displayed the strongest positive correlation with water quality variables (Table S6), most notably conductivity (Spearman's $\rho = 0.78 - 0.79$) and TON ($\rho = 0.62 - 0.72$). Moderate correlations with conductivity were recorded for *%taxa_high* and *%taxa_motile*. In addition, these response variables displayed moderate correlations with TON in spring. Other response variables did not display any clear correlations with water quality variables.

# DISCUSSION

## Changes in assemblage structure

Our results clearly indicated that benthic diatom assemblages at biomonitoring sites downstream of water supply reservoirs were significantly different from assemblages at control sites and that these differences extended beyond the scale of individual reservoirs. Our results are in keeping with findings in the wider literature, predominantly on individual dams (*Blinn, Truitt & Pickart, 1989*; *Uehlinger, Kawecka & Robinson, 2003*; *Wu et al., 2009*; *Cibils Martina, Principe & Gari, 2013*; *Gallo et al., 2015*; *Algarte et al., 2016*). Large-scale studies on the subject are rare, but two regional-scale Australian studies involving eight

water supply reservoirs found that diatom assemblages downstream of reservoirs differed from assemblages inhabiting upstream sites (*Growns, 1999*; *Growns & Growns, 2001*).

In most instances, the proportion of statistical variation ($R^2$) explained by variable SiteType in this study was small. Values were no higher than 3% at the national scale, with $R^2$ values at the regional scale being less than 16%. We found that the spatial variables Region and DCpair (i.e., the reservoir group a sample site belonged to) explained a greater proportion of the total statistical variation than SiteType. This demonstrated that there were substantial regional influences on diatom assemblages, potentially confounding general effects of water supply reservoirs. The regional signal was typically strongest at control sites, suggesting a homogenising effect of river impoundment on diatom assemblages at downstream sites. Although diatoms are generally characterised by a cosmopolitan distribution (*Stevenson, Pan & Van Dam, 2010*), several large-scale riverine diatom studies have highlighted that physiographical patterns in assemblage structure should be taken into account (*Potapova & Charles, 2002*; *Leira & Sabater, 2005*; *Tison et al., 2005*). Furthermore, we assume that part of the statistical variation not explained by SiteType comprised temporal effects. Although the Year variable was found to be largely insignificant, it is likely that the multi-annual nature of the dataset introduced some temporal variation in the analysis.

## Changes in taxa richness, ecological guilds and species

We found significantly more benthic diatom taxa at monitoring sites downstream of water supply reservoirs than at unregulated control sites at regional and national scales. Literature on diatom taxa richness in association with impoundment is relatively scarce, but most studies have reported a greater number of taxa at sites downstream of impoundments. A regional study in South East Australia found that taxa richness appeared slightly higher at regulated sites than at control sites, but the difference was not significant (*Growns, 1999*; *Growns & Growns, 2001*). A study on run-of-river dams in China demonstrated that downstream species richness was significantly higher than predicted (*Wu et al., 2010*), while a higher periphyton taxa richness (including diatoms) was recorded downstream of a hydropower dam in the Soa River, Slovenia (*Smolar-Žvanut & Mikoš, 2014*). In addition, a number of studies reported an increase of diatom or periphyton biomass downstream of impoundments (*Uehlinger, Kawecka & Robinson, 2003*; *Chester & Norris, 2006*; *Nichols et al., 2006*; *Gallo et al., 2015*).

The increase of taxa richness downstream of water supply reservoirs was recorded for all three guilds, but the greatest increase was observed in the motile taxa guild, resulting in a greater relative richness of these taxa at downstream sites. Very few studies have used ecological guilds to quantify the effect of impoundment on benthic diatom assemblages (e.g. *Algarte et al., 2016*). A study on the Achiras Stream (Argentina), using an alternative morphological guild classification, reported that the benthic diatom assemblage downstream of the reservoir comprised a higher percentage of late-succession species than upstream, which was linked to a reduced seasonality and variability of flow (*Cibils Martina, Principe & Gari, 2013*).

Although we expected that indicator taxa would broadly correspond with the guild classification, for example indicators at control sites being mainly from the low-profile rather than the high-profile guild, our results do not support this. Most indicator species within the low-profile guild were indicators of downstream sites, while the high-profile guild comprised both indicators of downstream and control sites. The motile guild comprised more indicators of downstream than control sites. A genus classification appeared to offer a better prediction of site type (downstream or control site), since indicator species within genera were typically confined to one site type. This suggests that diatom data at genus-level classification may be sufficient for studying the effect of abiotic drivers, depending on the research or management requirements, as proposed in several sources in the literature (*Growns, 1999*; *Wunsam, Cattaneo & Bourassa, 2002*; *Rimet & Bouchez, 2012a*).

## Possible drivers of diatom assemblage changes

Trophic Diatom Index (TDI) scores were generally higher at downstream sites than at control sites and were found to be markedly higher in the Midlands than any other region, indicating more eutrophic conditions. In addition, we found that some water quality variables, including conductivity, were slightly higher at downstream sites and that values for TON, $NH_3$, orthophosphate (OP) and conductivity were markedly higher in the Midlands region. Within the UK, TDI has been used to assess phytobenthic ecosystems (*Kelly et al., 2008*; *WFD-UKTAG, 2014*) following the implementation of the Water Framework Directive (WFD), which requires member states to ensure heavily modified waterbodies, including impoundments and regulated rivers downstream, achieve "Good ecological potential" (GEP) (*Acreman & Ferguson, 2010*). Diatoms are known to be responsive to changes in nutrient conditions, particularly with respect to phosphorus (P) and nitrogen (N), and this principle forms the basis for the TDI methodology (*Kelly & Whitton, 1995*). We recorded strong correlations between TDI and water quality variables including conductivity and TON, as well as moderate correlations between some water quality variables and the proportion of motile diatoms. A number of studies have demonstrated the association between benthic diatoms and water quality including conductivity and nutrient enrichment (*Passy et al., 2004*; *Leira & Sabater, 2005*; *Bergey, Desianti & Cooper, 2017*; *Dalu et al., 2017*). The motile guild primarily consists of eutrophic and pollution-tolerant species and is typical of nutrient-rich habitats, and the relationship between nutrient concentrations and diatom motility is well established in literature (e.g. *Passy, 2007*; *Lange et al., 2011*; *Stenger-Kovács et al., 2013*; *Jones et al., 2017*).

A number of studies involving benthic diatom assemblages or algal communities have related community changes to abiotic factors other than water quality. The relationship between diatoms and river discharge has been demonstrated (*Molloy, 1992*; *Passy, 2001*; *Neif et al., 2017*) and a number of studies on river regulation have related diatom assemblage changes to flow alteration (*Peterson, 1986*; *Growns, 1999*; *Cibils Martina, Principe & Gari, 2013*). Taxa within the low-profile guild are likely to thrive in frequently disturbed habitats, whereas high-profile taxa are typically prevalent in conditions with little physical disturbance (*Passy, 2007*; *Stenger-Kovács et al., 2013*). However, the wider literature is not unanimous on the relationship between the motile diatom guild and flow. Some studies

have linked motile taxa to lower discharges (*Passy, 2007*; *Dalu et al., 2017*), while others reported an association with higher discharge (*Stenger-Kovács et al., 2013*; *Wu et al., 2019*). We found a general increase of motile taxa downstream of reservoirs, as well as a substantial number of indicator taxa at downstream sites belonging to this guild, but more research is needed to disclose the role of flow alteration on this guild of diatoms. In addition, a higher relative abundance of planktic diatoms was recorded at downstream sites than at control sites, with the difference being most pronounced for autumn samples. We hypothesise that most planktic diatom taxa originated from the reservoir upstream, being washed downstream with flushing or spilling flows (*Köhler, 1994*). Relative abundance at downstream sites was higher in autumn than in spring, which was likely related to the increased production of phytoplankton in the reservoir over the summer period. However, the presence of planktic cells at control sites indicated that at least some of the downstream planktic diatoms may have had an autochthonous origin (*Growns & Growns, 2001*). More stable medium- to low-flow conditions following impoundment may have facilitated the development of a riverine planktic diatom assemblage (*Petts, 1984*) and resulted in the increases recorded at downstream sites, especially during the autumn. The influence of temperature on diatom assemblages has also been reported (*Bergey, Desianti & Cooper, 2017*), and thermal alteration by impoundment has been shown to induce changes to assemblage composition within a short time-period (*Blinn, Truitt & Pickart, 1989*). In addition, the relationship between diatom motility and fine sediment has been demonstrated, although some studies indicated that there may be confounding effects of other environmental pressures (*Jones et al., 2014*; *Neif et al., 2017*; *Jones et al., 2017*).

Our findings indicating increased TDI values, the proportion of motile diatoms, and water quality variability downstream of reservoirs suggest the primary influence of water quality gradients on benthic diatom assemblages, even in largely oligotrophic regions, and the influence of river impoundment on downstream water quality. Nevertheless, it would require a modelling approach, possibly combined with field or laboratory experiments ideally including multiple abiotic factors, to determine whether the reported statistical correlations imply direct cause–effect relationships or relate to other indirect processes downstream of reservoirs.

## Study implications

Various studies have emphasised the importance of widely applicable flow–ecology relationships to underpin studies of riverine ecosystems (including regulated systems) at larger spatial scales (*Webb et al., 2013*; *Bruckerhoff, Leasure & Magoulick, 2019*). The current study demonstrated consistent responses of benthic diatom assemblages to water supply reservoirs in England, but also highlighted that the predictive power at the largest spatial scales was comparatively low. Recent studies using riverine benthic diatom assemblages have applied the concept of functional and morphological traits including ecological guilds together with taxonomic assemblage composition (e.g. *Dong et al., 2016*; *Jamoneau et al., 2018*; *Sun et al., 2018*). Our results on diatom ecological guilds in association with water supply reservoirs indicate that morphological trait classifications can be a valuable tool in river impoundment studies beyond the site-specific scale. Future studies should consider

increasing the diversity of substrates from which benthic diatoms are sampled, as ecological guild distribution have been shown to be habitat-specific (*Passy, 2007*). The high-profile guild is known to favour rock substrates, which represented the main sampling habitat in many riverine diatom studies and potentially explained the comparatively high proportion of high-profile taxa recorded in the current study.

The large-scale effects of impoundment on riverine communities have often been studied using macroinvertebrate community and fisheries data (e.g. *Cooper et al., 2016*; *Krajenbrink et al., 2019*), with the effect of abiotic factors including hydrological and thermal alteration on these ecosystem groups being well demonstrated (e.g. *Armanini et al., 2014*; *White et al., 2017*). Given the reported sensitivity of benthic diatoms to water quality gradients, it may prove challenging to fully quantify the effects of flow and temperature changes downstream of reservoirs (*Dalu et al., 2017*). No large-scale hydrological and temperature datasets were available in the current study. Further research using detailed data on the abiotic factors flow and water temperature is required to be able to disentangle the specific effects of flow and thermal alteration on benthic diatom assemblages from background water quality drivers. In addition, data on sediment transport and turbidity may provide further insights into the drivers of motile diatom taxa downstream of reservoirs.

## CONCLUSIONS

This is one of the first studies to use a large-scale benthic diatom dataset from a routine biomonitoring network covering multiple years to compare diatom assemblages at sites downstream of water supply reservoirs with unregulated control sites. Diatom assemblage structure at downstream sites was consistently different from assemblage structure at paired control sites beyond the site-specific scale (research question 1). These generalisable effects may help water resource managers and researchers decide whether mitigation measures downstream of water supply reservoirs are necessary and whether these measures can be applied at multiple reservoir locations with similar properties. The effect of other spatio-temporal variables on diatom assemblage differences was evident, especially at larger spatial scales, but we demonstrated that water supply reservoirs dampened the regional influence patterns in downstream assemblages. This study is also one of the first to use diatom ecological guilds in association with the effects of water supply reservoirs and demonstrates their value in river impoundment studies. A generally higher number of diatom taxa was observed at downstream sites, with the strongest increases found within the motile guild. Finally, it was demonstrated that diatom assemblage data can be used to identify environmental gradients associated with water supply reservoirs (research question 2). Water quality variables were strongly correlated with diatom assemblages, but the influence of other abiotic factors, for example flow and temperature, could not be ruled out and should be investigated in future research to quantify their influence.

## ACKNOWLEDGEMENTS

The views expressed in this paper are those of the authors and not necessarily those of the Environment Agency of England. The authors would like to express their gratitude

towards the various Environment Agency teams for collecting, preparing and screening the SHEBAM ecological dataset, and towards Caroline Howarth for overseeing the data collection process. We would also like to thank Martyn Kelly, three anonymous reviewers, and the editor for their helpful and constructive comments that have greatly improved the clarity of the manuscript.

### Funding

This work was supported by the Natural Environment Research Council (NERC grant number NE/L002493/1). The funders had no role in study design, data collection and analysis, decision to publish, or preparation of the manuscript.

### Grant Disclosures

The following grant information was disclosed by the authors:
Natural Environment Research Council: NE/L002493/1.

### Competing Interests

Michael J. Dunbar and Libby Greenway are employed by the Environment Agency of England.

### Author Contributions

- Hendrik J. Krajenbrink conceived and designed the experiments, performed the experiments, analyzed the data, prepared figures and/or tables, authored or reviewed drafts of the paper, approved the final draft.
- Mike Acreman, Michael J. Dunbar, Libby Greenway, Cédric L.R. Laizé and David B. Ryves conceived and designed the experiments, performed the experiments, authored or reviewed drafts of the paper, approved the final draft.
- David M. Hannah and Paul J. Wood conceived and designed the experiments, authored or reviewed drafts of the paper, approved the final draft.

### Data Availability

The raw data is available in a Supplementary File.

### Supplemental Information

Supplemental information for this article can be found online at http://dx.doi.org/10.7717/peerj.8092#supplemental-information.

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
