# Peer review of "Diatoms as indicators of the effects of river impoundment at multiple spatial scales"

_PeerJ, doi:10.7717/peerj.8092_

## Round 0.1 · original submission · Minor Revisions

This manuscript provides new findings about diatom in river impoundment at different scales. The results are interesting. However, more interpretation of the results are needed. More description on the diatom ecology and species are required. Also more references about individual species are needed, particularly in Discussion

Reviewer 1 ·

Basic reporting

No comment

Experimental design

No comment

Validity of the findings

No comment

Additional comments

Comments on "Benthic diatoms as indicators of river impoundment at multiple spatial scales"
I have no real comment on the study design, data collection or analyses as the two former are part of a national standard and the multivariate and univariate analyses are common-place for ecological data. There are a few statements in the text that need explanation and I have a few additional comments, as follows.
1. The title needs rethinking as planktonic diatoms are included in the data analysis, results and discussion
2. Line 101 page 3 – "Only monitoring sites with sufficiently good water quality…" More details are required for this. How did the authors define good water quality? What parameters were used? The authors find that water quality is an important driver of the diatoms even with "sufficiently good water quality", so the data set is already confounded, so why not include those with less quality?
3. Page 4 Community structure – Is it possible for the authors to get access to the Primer 7 software package in order to do a bootstrapped ordination. Only bootstrapped PCAs are currently available in R and probably would not be appropriate for the diatom data. Unless the authors have sufficient programming skills to write a bootstrapped mds? A mds bootstrapped ordination would be extremely useful to look at the entire data set in one diagram, rather than separating all the samples into the two seasons and having multiple ordinations for the different regions.
4. Page 4 Diatom indices – The authors need to provide hypotheses as to why the different ecological guilds would react differently to river regulation, i.e. provide an explanation why they were grouped in such a way and analysed separately.
5. Lines 231 and 232 page 6 – "on average higher at downstream sites than control sites" is repeated in the following line "Ntaxa was typically higher at downstream sites". This paragraph need rewording.
6. Tables 3 and 4 – Can the authors provide a percentage variance explained as in Tables 1 and 2?
7. Lines 403 and 404 page 10 – The term "may guide water managers" is very vague. The authors need to be much clearer about what that means and then the paper would be more valuable to the scientific and management communities. For example, some managers (and maybe conservationists) would see it as a good thing that there are more taxa at the downstream sites as there is more biodiversity.
8. Line 412 page 11 – "suggested the influence of other factors including flow regime". With no hydrological data this statement cannot be made.

Reviewer 2 ·

Basic reporting

This study presents a big dataset of diatom data to reveal the differences exist between diatom communities at sampling sites downstream of water supply reservoirs and unregulated control sites at the regional and national scale. I have noticed that the authors have published the paper Krajenbrink et al. (2019, STOTEN) which also is based on the same dataset from the SHEBAM monitoring network. This manuscript answers questions similar as those in the 2019 STOTEN paper from the angle of diatom communities. I think the conclusions raised here are of high scientific validity and I am happy to see some discussion on the connections between the two research (i.e. ecological difference between diatom and macroinvertebrate communities). The current ms is written in clear and professional English and all the statistic analysis are sound and appropriately used.

Experimental design

Well designed and rigorous performed

Validity of the findings

Conclusions are well stated and scientificly justified.

Additional comments

I have only one comment. Is it possible to give more description on the diatom ecology and species (i.e. the dominant species from downstream and control sites?

·

Basic reporting

All good. One final read through in order to prune back surplus wordage and check for clarity would be useful.

Experimental design

No substantial issues here. Meets the journal's requirements

Validity of the findings

Some suggestions to the authors are included below. These relate primarily to data interpretation, not the execution of the work per se. Some speculation in the Discussion is not evidence-based and could be pruned back

Additional comments

I am happy to recommend this paper for publication, subject to a few reservations discussed below. It is a generally well-written contribution to the literature on this topic, and is based around thorough statistical analysis of a large dataset. My primary reservation about the paper is summed up by the complete absence of reference to individual diatom species which is also reflected in the lack of profound insight in the Discussion. It is, nonetheless, a workmanlike paper that does the job it sets out to do.

My own experience is that the phytobenthos in rivers immediately below impoundments often take on the characteristics of the phytobenthos in the impoundment itself. This is true, to some extent, for both natural lakes and for artificial water bodies and probably is due to inocula from the standing water body but also to the nature of the regulation itself. Inclusion of samples from upstream and downstream of natural lakes would have made the paper more interesting, as would inclusion of samples from within the reservoirs.

I disagree with some of the data interpretation presented here. First, I am not convinced that biogeographical factors are responsible for differences at the national scale. Second, the reasons for the observed changes in the downstream assemblages need a rethink.

The experience in the UK is that, in the absence of anthropogenic pressure, it is alkalinity (reflecting underlying geology) that is the primary factor controlling diatom assemblages. I suspect that the differences you see between northern and southern England are probably a consequence of underlying geology and I would not regard this as “biogeography” per se, simply as a shuffling of environmental drivers that happens to have a spatial correlation. I’m happier when you refer to this as “regional influence”

Lines 344-347 say “The results involving TDI and the motile guild point towards the influence of water quality gradients on benthic diatom communities, even in largely oligotrophic regions, and provide evidence for the influence of river impoundment on downstream water quality.” This is an easy default interpretation but does your data actually support this? Table S4 and associated figures suggest that N variables have a larger effect than OP whereas dogma would suggest that it is OP that should be driving these changes. First, are there reasonable grounds to argue that N is limiting, relative to P? Second, are there unmeasured variables that may correlate with these that are actually driving the patterns you observe? A further possibility is that the biological changes are only partially captured by the diatom assemblage – that there is greater accumulation of biomass in regulated rivers immediately downstream of impoundments (possibly due to filamentous greens accumulating due to the less scouring conditions) and that this favours certain diatom taxa that also happen to thrive when the thick biofilms are the result of eutrophication. I don’t have an easy answer either, but I think that, in trying to explain your results, you need to acknowledge that there is not a direct cause-effect relationship between impoundments and diatoms and that much of the evidence on which you rely for this part of the discussion is also based on statistical associations within datasets rather than experiments. In the absence of a well-thought through conceptual model to explain the effects observed, the easiest way forward may simply be to trim back some of the speculation in this part of the Discussion. I certainly find lines 369-371 rather strong, in light of the evidence. Perhaps “most important drivers …” should be replaced by “strongest correlates”?

Minor comments
Please refer to the diatom “assemblage”: the diatoms are part of a wider phytobenthic/periphytic community

Line 128-129: what are “EA quality assurance protocols”. I did not know that they had a standardised process for this.

Why did you use TDI3 when TDI4 is the EA’s current approach?

To what extent is Ntaxa affected by the counting policy described in lines 123-124? If there is an overwhelmingly dominant taxon then the “tail” of rare species will be correspondingly longer due to the greater count size and Ntaxa for such samples may be misleading.

It would be useful to have a Table similar to S5 that describes the inter-relationships between the diatom indices that you calculate, and a comment on the extent to which each is contributing unique information.

Reviewer 4 ·

Basic reporting

See comments below

Experimental design

See comments below

Validity of the findings

See comments below

Additional comments

General comments:
This study test if diatom community, taxonomic diversity, diatom guilds etc. show consistent response to impoundment with data from multiple scales. Overall, the author did a good work. But I also have some comments and suggestions.
1) Line 86-91, the author clarified their research question, which is fine. But I would suggest to include some expectations/hypothesis, e.g., what does author expect diatom community, diversity etc. differ among control VS impacted sites? Are these show consistency in different region?
2) Line 274-278, for exploring the relationships between water quality and benthic algal responses, the author only employed a correlation analysis, which in my opinion is not sufficient. With such results, we are not sure if these correlations are consistent in different regions, i.e., for some indices, the main driver will be different. Therefore, I would recommend a linear mixed model for this part, that is, take regions and season as random effects, while water quality variables as fixed effects.
3) Line 323-325, author stated that higher TDI in Midlands indicated more eutrophic conditions. I am wondering if nutrient concentration support this?
4) Line 352-354, the author said that motile and high-profile taxa are typically prevalent in low disturbance conditions. This is not right, only high-profile taxa adapt to low disturbance, see
(Wu et al., 2019). This also means Line 354-355 is wrong, please check below references.
Wu, N., Thodsen, H., Andersen, H.E., Tornbjerg, H., Baattrup-Pedersen, A., Riis, T., 2019. Flow regimes filter species traits of benthic diatom communities and modify the functional features of lowland streams - a nationwide scale study. Sci. Total Environ. 651, 357–366. https://doi.org/10.1016/j.scitotenv.2018.09.210
5) Suggestions for visualization. The author showed their results mainly with tables, which is fine but not very intuitive. Especially for Table 3 and Table 4, I noticed that author do have plots for these two tables in supplementary, so please do the reverse way. In addition, for the plots, significance can be indicated with letters. As for tables, more details like standard deviation, chi-square value for Kruskal-Wallis analysis should be included.

---

## Round 0.2 · accepted · Accept

You have answered reviewers’ questions and made large improvement of ms. The ms it is much clearer now.